# Three-Dimensional Nondestructive Isotope-Selective Tomographic Imaging of $^{208}$Pb Distribution via Nuclear Resonance Fluorescence

**Khaled Ali** [1,2,*], **Heishun Zen** [1], **Hideaki Ohgaki** [1], **Toshiteru Kii** [1], **Takehito Hayakawa** [3,4], **Toshiyuki Shizuma** [3], **Hiroyuki Toyokawa** [5], **Masaki Fujimoto** [6], **Yoshitaka Taira** [6] and **Masahiro Katoh** [6,7]

1   Institute of Advanced Energy (IAE), Kyoto University, Gokasho, Uji, Kyoto 611-0011, Japan; zen@iae.kyoto-u.ac.jp (H.Z.); ohgaki.hideaki.2w@kyoto-u.ac.jp (H.O.); kii@iae.kyoto-u.ac.jp (T.K.)
2   Physics Department, Faculty of Science, South Valley University, Qena 83523, Egypt
3   Tokai Quantum Beam Science Center, National Institutes for Quantum and Radiological Science and Technology (QST), Ibaraki 319-1106, Japan; hayakawa.takehito@qst.go.jp (T.H.); shizuma.toshiyuki@qst.go.jp (T.S.)
4   Institute of Laser Engineering, Osaka University, Suita, Osaka 565-0871, Japan
5   Institute of Advanced Industrial Science and Technology (AIST), Tsukuba Central 2-4, Ibaraki 305-8568, Japan; h.toyokawa@aist.go.jp
6   UVSOR-III Synchrotron Facility, Institute for Molecular Science, National Institutes of Natural Sciences, Okazaki 444-8585, Japan; mfmoto@ims.ac.jp (M.F.); yostaira@ims.ac.jp (Y.T.); mkatoh@ims.ac.jp (M.K.)
7   Hiroshima Synchrotron Radiation Center, Hiroshima University, Higashi-Hiroshima 739-0046, Japan
*   Correspondence: khaled.ali.28e@st.kyoto-u.ac.jp; Tel.: +81-774-38-3425; Fax: +81-774-38-3426

**Abstract:** Combining the nuclear resonance fluorescence (NRF) transmission method with computed tomography (CT) can be a novel method for imaging the isotope distributions, which is indispensable in nuclear engineering. We performed an experiment to reconstruct a three-dimensional NRF-CT image with isotope selectivity of enriched lead isotope rods ($^{208}$Pb) together with a set of different rods, including another enriched isotope ($^{206}$Pb), iron, and aluminum rods, inserted into a cylindrical aluminum holder. Using a laser Compton scattering (LCS) gamma ray beam with a 5.528 MeV maximum energy, 2 mm beam size, and 10 photon·s$^{-1}$·eV$^{-1}$ flux density, which is available at the BL1U beamline in the ultraviolet synchrotron orbital radiation-III (UVSOR-III) synchrotron radiation facility at the Institute of Molecular Science at the National Institutes of Natural Sciences in Japan, and we excited the J$^{\pi}$ = 1$^{-}$ NRF level at 5.512 MeV in $^{208}$Pb. An isotope-selective three-dimensional NRF-CT image of the $^{208}$Pb isotope distribution was experimentally obtained for the first time with a pixel resolution of 4 mm in the horizontal plane.

**Keywords:** gamma rays; laser Compton scattering (LCS); nondestructive inspection; nuclear resonance fluorescence (NRF); isotope-selective computed tomography (CT)

## 1. Introduction

The use of the nuclear resonance fluorescence (NRF) in combination with computed tomography (CT) has been demonstrated as an isotope imaging technique, allowing for nondestructive inspection (NDI) for detecting hidden isotopic compositions of target materials that are of interest [1]. Furthermore, the isotope selectivity assessment [2] is a significant advancement in nuclear engineering. Physically, NRF is a nuclear process in which a nuclear state is excited through the absorption of a photon whose energy is nearly identical to the excited energy of the nuclear state. Subsequently, the excited state decays back to its initial state by emitting a photon or a cascade of photons. By measuring the emitted photon energy, a specific nuclear isotope of interest can be identified [1–9]. NRF can be used to investigate for commercial contraband, explosives, and special nuclear materials (SNMs) [3,10,11]. The promise of the NRF technique, which has isotope sensitivity, as an

NDI technique in safeguard applications lies in its ability to directly measure a specific isotope in an assay target without having to unfold the combined responses of several fissile isotopes, as is often required by other NDI methods [12,13]. Because of its high penetrability, an MeV energy region monochromatic gamma ray is an effective probe for the NDI of high-density materials. In NRF measurements, laser Compton scattering (LCS) gamma rays [14] are used extensively to excite the nuclei to be detected because of their characteristics, such as directivity, a quasi-monochromatic energy spectrum, tunable energy, controllable polarization, and high flux [15]. Kikuzawa et al. [6] performed a proof-of-principle experiment for NRF nondestructive analysis with LCS gamma ray beams. A lead isotope of $^{208}$Pb inside of an iron box was measured as a high-Z material instead of fissionable isotopes.

CT is one of the cross-sectional imaging modalities which has numerous applications in various fields, such as medical imaging. CT systems are also used for checking luggage at airports worldwide to detect explosives [16–19]. For imaging isotopes inside massive materials, the integration of CT and NRF (NRF-CT) has been proposed [20]. Daito et al. [21] employed a GEANT4 Monte Carlo simulation code to perform a simulation study of the CT imaging of nuclear distribution by NRF processes induced by quasi-monoenergetic LCS gamma rays. Recently, we carried out a proof-of-experiment for the NRF-CT using an LCS gamma ray beam available at the beamline BL1U in the ultraviolet synchrotron orbital radiation-III (UVSOR-III) synchrotron radiation facility at the Institute of Molecular Science at the National Institutes of Natural Sciences in Japan [15]. We measured a two-dimensional (2D) NRF-CT image of a natural lead isotope ($^{208}$Pb) implanted in an aluminum sample based on the transmission method (notch detection method) [15]. One of the advantages of this technique is the ability to measure the CT image of an isotope of interest. Therefore, the isotope-selective capability was demonstrated using an LCS gamma ray beam for enriched lead isotope rods ($^{208}$Pb and $^{206}$Pb) [2]. The reconstructed image had a resolution of 2 mm in pixel size. This result suggests that when a three-dimensional visualization of a specific isotope within an assay volume is provided, we get the full perception of the hidden isotopes. Since the demonstrated 2D NRF-CT imaging required approximately 60 h of acquisition time [2], we should clarify the limitation of the NRF-CT technique with the LCS gamma ray beam to study a three-dimensional (3D) NRF-CT image under the current experimental conditions and improve the measurement and analysis procedures accordingly.

In the present study, we demonstrate a 3D NRF-CT image with three layers of 2D NRF-CT images within the same acquisition time or less than the previous study [2]. To accomplish this, some adjustments in the experimental setup from the previous study were performed [2], including doubling the flux intensity of the LCS gamma ray beam and reducing the scanning steps number for the images in two dimensions. Consequently, we reconstructed a 3D NRF-CT image of enriched lead isotope rods ($^{208}$Pb) inserted with different rods ($^{206}$Pb, iron, aluminum, and a vacancy) inside a thick aluminum holder, based on the transmission method.

## 2. Experimental Procedure

The experimental setup was similar to that of a previous study [2], which was performed at the BL1U beamline of the UVSOR-III synchrotron radiation facility [22,23]. Figure 1 displays the schematic diagram of the experimental setup. The LCS gamma rays were generated by a head-on collision between high energy electrons in the UVSOR-III storage ring and a laser beam via Compton scattering. The electron energy in the storage ring was $746 \pm 1$ MeV, and the average beam current was 300 mA in the top-up mode [22]. A continuous wave (CW) Tm-fiber laser system (TLR-50-AC-Y14, IPG Photonics GmbH, Köttinger Weg 188 Wissen, Germany) with a lasing wavelength of 1.896 μm (0.6539 eV) and a maximum average power of 50 W was used as the laser source. The laser beam was randomly polarized. In this setup, the maximum gamma ray energy and the total flux with a 100% energy bandwidth were 5528 keV and $10^8$ photons·s$^{-1}$, respectively. The maximum

energy of the LCS gamma ray beam was able to excite the state $J^\pi = 1^-$ at 5.512 MeV in $^{208}$Pb [24].

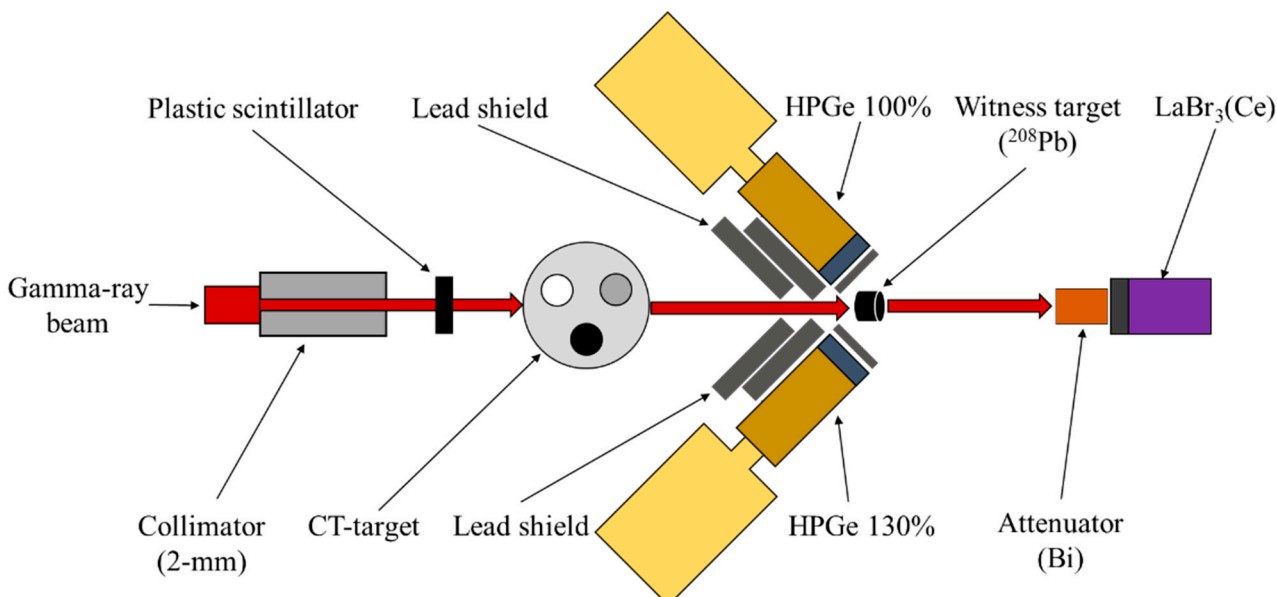

**Figure 1.** Schematic diagram of the experimental setup at the BL1U beamline in the ultraviolet synchrotron orbital radiation-III (UVSOR-III) synchrotron radiation facility. A lead collimator is positioned at the laser Compton scattering (LCS) gamma ray beam path. A plastic scintillator is used to measure the flux of the incident LCS gamma ray beam. The LCS gamma ray beam path is intercepted by a computed tomography (CT) target, which is positioned on a three-dimensional moving stage. Two high-purity germanium (HPGe) detectors, shielded by lead blocks, are used to measure the nuclear resonance fluorescence (NRF) gamma rays scattered from a witness target ($^{208}$Pb). To prevent pile-up events, a bismuth absorber is used in front of an LaBr$_3$(Ce) scintillation detector, which measures the transmitted gamma rays.

To define the LCS beam diameter and the energy spectrum on a CT target, a lead collimator (20 cm × 10 cm × 10 cm) with a 2 mm internal hole diameter was positioned at the LCS gamma ray beam path. As was numerically simulated using the EGS5 Monte Carlo simulation code [25], the gamma ray flux density passing through the collimator was found to be 10 photons·s$^{-1}$·eV$^{-1}$ at the resonance energy (5.512 MeV). In this experiment, the gamma ray flux density was double that of our previous study, because we used a collimator with a hole diameter twice that of the previous setup [2]. A 5 mm thick plastic scintillation detector (OHYO KOKEN KOGYO Co., Ltd. Kumagawa, Fussa City, Tokyo, Japan), which was located 160 cm downstream from the lead collimator, was used to measure the flux of the incident LCS gamma ray beam. After traversing the plastic scintillation detector, the transmitted LCS gamma ray beam was injected into the CT target. Subsequently, the LCS gamma ray beam transmitted from the CT target irradiated a witness target made of an enriched lead isotope rod ($^{208}$Pb) with a diameter of 6 mm and a length of 6 mm, which was positioned 65 cm away from the CT target. Two high-purity germanium (HPGe) detectors with efficiencies of 100% (AMETEK ORTEC, Tennessee, USA) and 130% (Mirion Technologies (CANBERRA), Connecticut, USA), relative to a 3" × 3" NaI (Tl) scintillator, were arranged at two positions with an angle of 120° with respect to the gamma ray beam axis and at a vertical distance of 8 mm from the witness target. The HPGe detectors were used to measure the gamma rays scattered from the witness target ($^{208}$Pb). A 6 mm thick lead absorber plate was installed between each HPGe detector and the witness target. Moreover, thick lead shields were positioned around the gamma ray beam axis in the arrangement to shield the HPGe detectors. The lead shields were helpful in preventing photon accumulation on the HPGe detectors and reducing the detector's dead time. To evaluate the atomic attenuation in the CT target, we measured the flux of the gamma rays that were transmitted from the witness target by a 3.5" × 4" LaBr$_3$(Ce) scintillation

detector (SAINT-GOBAIN, Courbevoie, France), which was placed 40 cm downstream from the witness target. A 10 cm thick bismuth absorber was mounted in front of the LaBr$_3$(Ce) detector to prevent signal pile-up. Although the bismuth absorber attenuated the gamma rays transmitted from the witness target, this attenuation was annulled during image reconstruction since the relative change of the gamma ray flux, depending on the CT target position and angle, was used in the analysis. The signals from each detector were independently recorded by a multichannel analyzer (APG7400, Techno AP Co., Ltd., Mawatari, Hitachinaka-shi, Ibaraki, Japan).

Figure 2a displays a picture of the CT target. A cylindrical aluminum holder with a height of 20 mm and a diameter of 25 mm was used as the target container, as shown in Figure 2b. Three holes with equal diameters of 6.1 mm were drilled into the holder body with a 120° pitch angle. To construct clear 3D images, the sample holder holes were filled by a set of cylindrical rods with equal dimensions of 6 mm diameter and height, as shown in Figure 2c. In hole 1, the bottom hollow was filled by an iron rod, followed by one of the enriched lead isotope rods ($^{208}$Pb), while the third hollow remained empty. In hole 2, the bottom was filled by one of the enriched lead isotope rods ($^{206}$Pb), followed by a hollow aluminum rod and a lead isotope rod ($^{208}$Pb). In hole 3, the bottom was filled by an enriched lead isotope rod ($^{208}$Pb), followed by an iron rod and an enriched lead isotope rod ($^{206}$Pb) in the hole top.

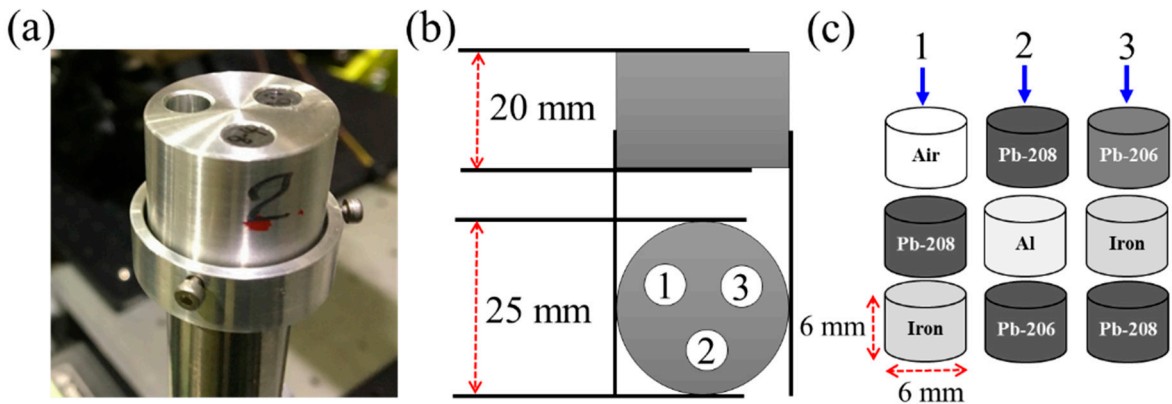

**Figure 2.** (**a**) Picture of the CT target. (**b**) Geometry of the CT target holder, having a diameter of 25 mm and a height of 20 mm. The target holder has three holes with equal diameters of 6.1 mm and a pitch angle of 120°. The numbering of the holes is shown on the target holder. (**c**) Arrangement of the rods, including the enriched lead isotope rods ($^{206}$Pb and $^{208}$Pb), iron, aluminum, and a vacant area.

The CT target was positioned 214 cm downstream from the collimator on a three-axis moving stage which could scan in three directions: vertical (*z*), rotational (*θ*), and horizontal (*x*). Figure 3 displays the scanning directions of the CT target. We chose the smallest number of the measurement positions that would allow us to obtain a reasonable image in an appropriate time frame and prove the NRF nondestructive analysis in three dimensions using the LCS gamma ray beam. The step sizes in individual directions were determined to obtain a reasonable image resolution, considering the CT target geometry under a limited measurement time. The diameter of the gamma ray beam was 2 mm, which was half the pixel size of the reconstructed image. The CT target had a diameter of 25 mm, and the lead and iron rods in the target had a diameter of 6 mm. The gamma ray beam diameter was sufficiently shorter than the object being studied. Three layers at *z* = 3, 11, and 17 mm from the holder bottom in the *z* direction were selected for constructing 2D NRF-CT images. The vertical positions of these layers were selected to obtain a 2D NRF-CT image for each horizontal row of the rods, since there were three rows of the rods in the *z* direction, as shown in Figure 2c. The CT target was rotated around the *θ* axis, with an angle step of 30° in the range from 30° to 180°. The scanning was also carried out in the *x* direction, with a step size of 4 mm in the range from −12 to +12 mm. We collected 126 data

points with the CT target (7 steps along the *x*-axis × 6 rotational angles ($\theta$) × 3 layers in the *z* direction). In addition, one data point for each horizontal scan in the *x* direction was measured for collecting data in the absence of the CT target. Therefore, a total of 144 points were measured. The averaged measurement time for each data point was approximately 20 min.

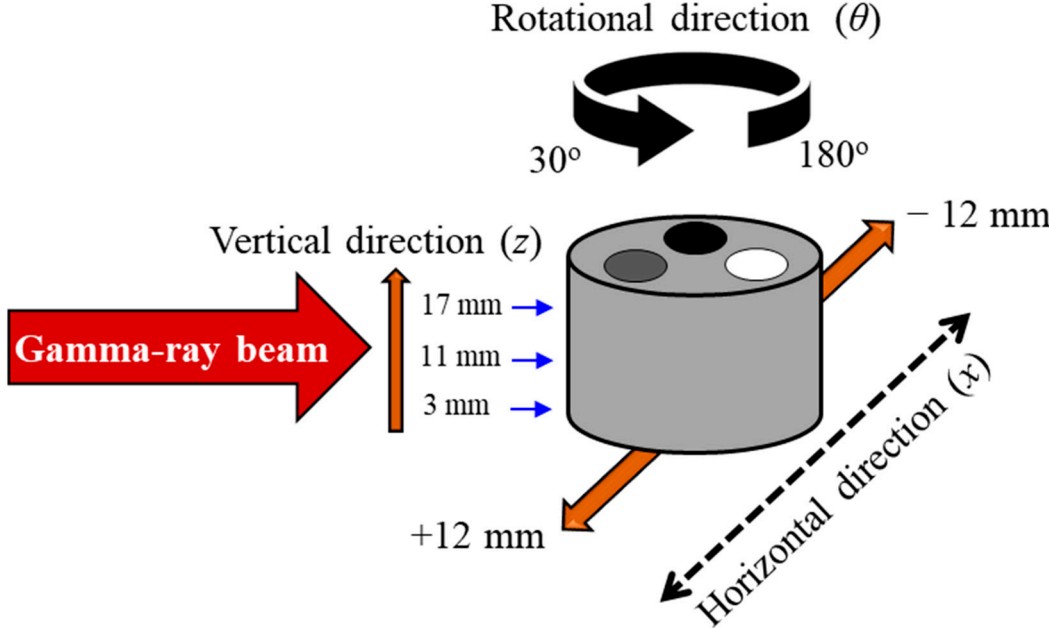

**Figure 3.** Data acquisition geometry of the CT target in three dimensions. In the horizontal direction (*x*), the CT target position was scanned from −12 mm to +12 mm in steps of 4 mm within the movement range. In the rotational direction ($\theta$), the CT target was rotated around the axis of rotation, with angle steps of 30° in the range from 30° to 180°. The three layers in the vertical direction (*z*) were scanned at 3, 11, and 17 mm from the CT target bottom.

To perform the experiment in three dimensions, we established an automated scanning system, which was programmed using LabVIEW software (National Instruments, Austin, TX, USA). The scanning system included the CT target motion control of the moving stage in the three dimensions (*z*, $\theta$, and *x*), data analysis, and the detector spectra recording.

### 3. Results and Discussion

The measurements of the gamma ray transmission factors, by including the effects of atomic absorption and nuclear resonance attenuation, yielded 2D NRF-CT images of the $^{208}$Pb rods for each layer in the *z* direction [2]. The 2D NRF-CT images were visualized together to obtain reconstructed 3D NRF-CT images. Figure 4 shows a typical gamma ray spectrum obtained from the plastic scintillation detector. The plastic scintillation detector had no energy resolution within the 5 MeV region. Thus, to obtain the relative flux of the incident LCS gamma ray beam, the region of interest (ROI) was chosen in a wide channel range from 50 to 7000.

Figure 5 displays a typical energy spectrum measured by the LaBr$_3$(Ce) scintillation detector. The energy resolution of the LaBr$_3$(Ce) detector was sufficiently high to distinguish between the LCS gamma ray peak and its single escape peak. The ROI, indicated in the shaded area, covered the energy range from 5360 keV to 5885 keV to consider the counts under the full energy peak.

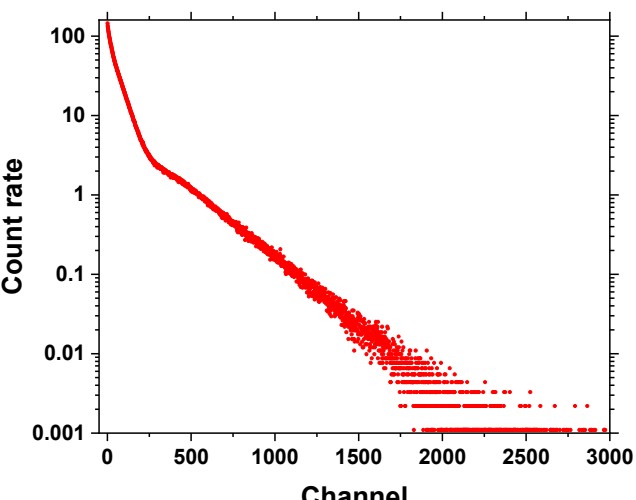

**Figure 4.** Part of the typical spectrum from the plastic scintillator (0:3000 channel). A few events were found in the channels ranging from 3001 to 8192.

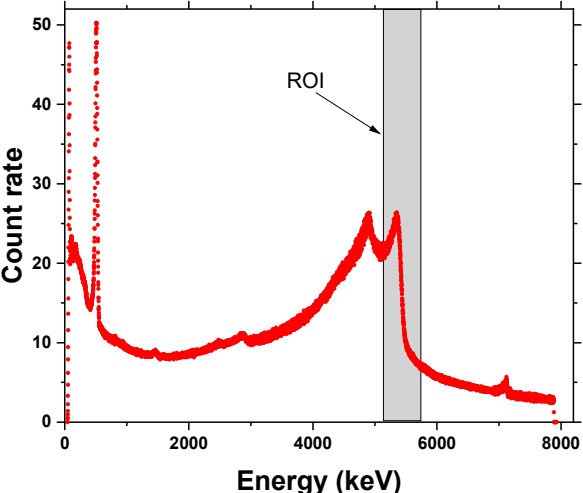

**Figure 5.** Typical LCS gamma ray spectrum obtained from the LaBr$_3$(Ce) detector. The shaded area is the region of interest (ROI).

For each point of measurement, we calculated the count rates of the plastic scintillator and the LaBr$_3$(Ce) detector by integrating the counts in the ROIs and dividing them by the acquisition time. Then, we calculated the transmission factor of the off-resonance gamma rays ($\varepsilon_{\text{off}}$) for each $\theta$ and $x$ of the CT target. To calculate the normalized count rate of the transmission detector, we used the same procedure as that of previous studies [15,21]. The count rate of the LaBr$_3$(Ce) detector was divided by the total integrated count rate of the plastic scintillator within the ROI for each target position. Then, at each data point, $\varepsilon_{\text{off}}$ was calculated by the normalized count rate of the transmission detector in the presence and absence of the CT target according to the following formulas:

$$\varepsilon_{\text{off}} = e^{\left[-\left(\frac{\mu}{\rho}\right)_{\text{ave}} \times \rho_{\text{ave}} \times L\right]} \tag{1}$$

$$\varepsilon_{\text{off}} = \left[\frac{c_{\text{off}}(\text{with CT target})}{c_{\text{off}}(\text{without CT target})}\right] \tag{2}$$

where $\left(\frac{\mu}{\rho}\right)_{\text{ave}}$ is the average mass attenuation coefficient, $\rho_{\text{ave}}$ is the average density, and $L$ is the length of the path through the CT target. The resonance attenuation effect on the yield measured by the LaBr$_3$(Ce) detector was negligible, as the width of the ROI was

wider than that of the resonance attenuation by 4–5 orders of magnitude. Figure 6 shows a typical summed energy spectrum measured by the HPGe detectors. There were several peaks in the HPGe spectrum. Two NRF levels of $^{208}$Pb appeared at the energies of 5512 keV and 5292 keV. The single and double escapes (S.E. and D.E.) of the $^{208}$Pb NRF at 5512 keV were represented by two peaks at the energies of 5001 keV and 4490 keV, respectively. The peaks at 2614 keV and 1460 keV originated from the natural unstable isotopes of $^{208}$Tl and $^{40}$K, respectively. Energy calibration of the detectors was performed using the 1460 keV peak of $^{40}$K and the 5512 keV NRF peak of $^{208}$Pb. The spectra measured by the two HPGe detectors were summed after the 2 keV per channel rebinning of the energy axis.

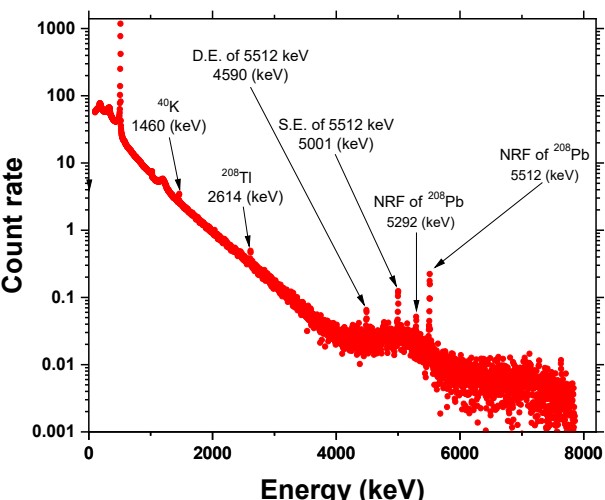

**Figure 6.** Typical summed spectrum of the two HPGe detectors with 2 keV rebinning.

The NRF peak at 5512 keV was clearly seen in the summed spectra of the HPGe detectors (Figure 6). As shown in Figure 7, the observed NRF peak was fitted with a Gaussian function using the least squares method. The area of the fitted Gaussian function was used as the NRF yield.

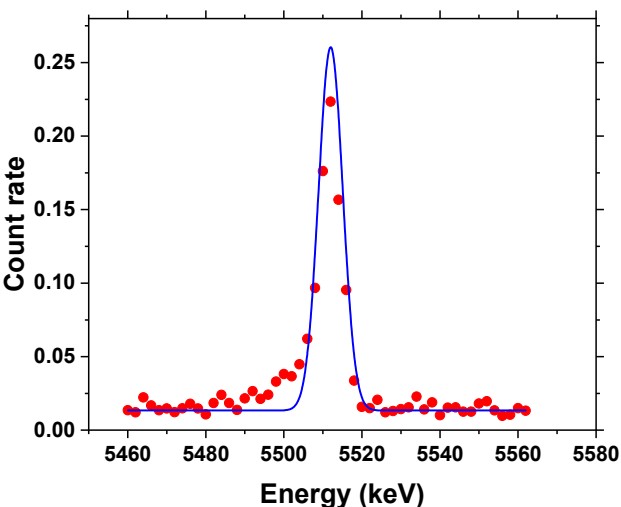

**Figure 7.** Gaussian fitted NRF peak of $^{208}$Pb at 5512 keV.

The normalized NRF count rate was calculated by dividing the NRF yield by the acquisition time and count rate of the plastic scintillator detector. Next, we calculated the transmission factor at the resonance energy ($\varepsilon_{on}$) for each $\theta$ and $x$ of the CT target. $\varepsilon_{on}$ was

calculated by the normalized NRF count rate obtained in the presence and absence of the CT target according to the following formulas:

$$\varepsilon_{\text{on}} = e^{\left[-\left(\left(\frac{\mu}{\rho}\right)_{\text{ave}} \times \rho_{\text{ave}} \times L + \sigma_{\text{NRF}} \times N_{\text{t}}\right)\right]} \tag{3}$$

$$\varepsilon_{\text{on}} = \left[\frac{c_{\text{on}}(\text{with CT target})}{c_{\text{on}}(\text{without CT target})}\right] \tag{4}$$

where $\sigma_{\text{NRF}}$ is the $^{208}$Pb NRF cross section and $N_{\text{t}}$ is the areal density of the nuclide in the CT target in the gamma ray beam path. We used the same procedure of calculations as that of previous studies (see [2,15,21]). The nuclear resonance attenuation factor $\varepsilon_{\text{NRF}}$ was obtained by the following equation [2,15]:

$$-\ln(\varepsilon_{\text{NRF}}) = -[\ln(\varepsilon_{\text{on}}) - \ln(\varepsilon_{\text{off}})] \tag{5}$$

As the sign-inverted natural logarithm of the attenuation factor ($-\ln(\varepsilon)$) is proportional to the number of isotope or scattering sources in the CT target, the images in 2D-CT and 3D-CT are reconstructed using the sign-inverted natural logarithm of the attenuations ($-\ln(\varepsilon_{\text{NRF}})$, $-\ln(\varepsilon_{\text{on}})$, and $-\ln(\varepsilon_{\text{off}})$).

In the NRF-CT imaging, several CT reconstruction algorithms could be used. Due to the limited number of the measured projections in the current study, one of the iterative algorithms, called the algebraic reconstruction technique (ART) [26–28], was used. A program was developed in LabVIEW to reconstruct the images. The ART is a sequential approximation method for image reconstruction from a series of angular projections (sinogram). To perform the ART algorithm, the sinogram data points were to be processed in a series of steps. We created a two-dimensional primary image as a two-dimensional matrix with zero in every cells. The cell number of the matrix was determined based on the projection numbers obtained during the experiment. The primary matrix in our case was $7 \times 7$. To mitigate the errors of the matrix rotation caused by the small matrix size of the sinogram, pre- and post-processing were implemented. The difference between the projections calculated from the reconstructed image and the measured projections was computed and used to correct the reconstructed image iteratively. The convergence of the reconstruction was evaluated by the root-mean-square difference between the previous and current reconstructed images. More details of the reconstruction process are provided in [15].

For each $z$ layer, three reconstructed images for off-resonance, on-resonance and pure NRF attenuations were obtained with a 4 mm pixel size. Figure 8a shows a cross-sectional slice image of the CT target arrangement at $z = 3$ mm from the CT target bottom. This shows that $^{206}$Pb and $^{208}$Pb rods were inserted into the first two holes of the aluminum holder, whereas an iron rod was inserted into the third hole. The CT reconstructed image of the off-resonance attenuation, measured by the LaBr$_3$(Ce) detector, is shown in Figure 8b. The two high-attenuation areas that were induced by the atomic process corresponding to the $^{206}$Pb and $^{208}$Pb rods were clearly visible in the reconstructed image. However, it was impossible to distinguish between these two isotope rods, because the atomic attenuations by $^{208}$Pb and $^{206}$Pb were almost the same. In comparison with these lead rods, the atomic attenuation caused by the iron rod was significantly less. Thus, it was difficult to see the iron rod in the reconstructed image. Figure 8c shows the distribution of the on-resonance gamma ray attenuation. Although the $^{208}$Pb rod was clearly seen, the intensity of the $^{206}$Pb rod was comparatively less. Moreover, the iron rod almost vanished. The distribution of NRF attenuation by $^{208}$Pb (pure NRF) after subtraction of the atomic attenuation is shown in Figure 8d. Although the $^{208}$Pb rod was clearly visible, both the other lead-enriched isotope ($^{206}$Pb) rod and iron rod almost vanished.

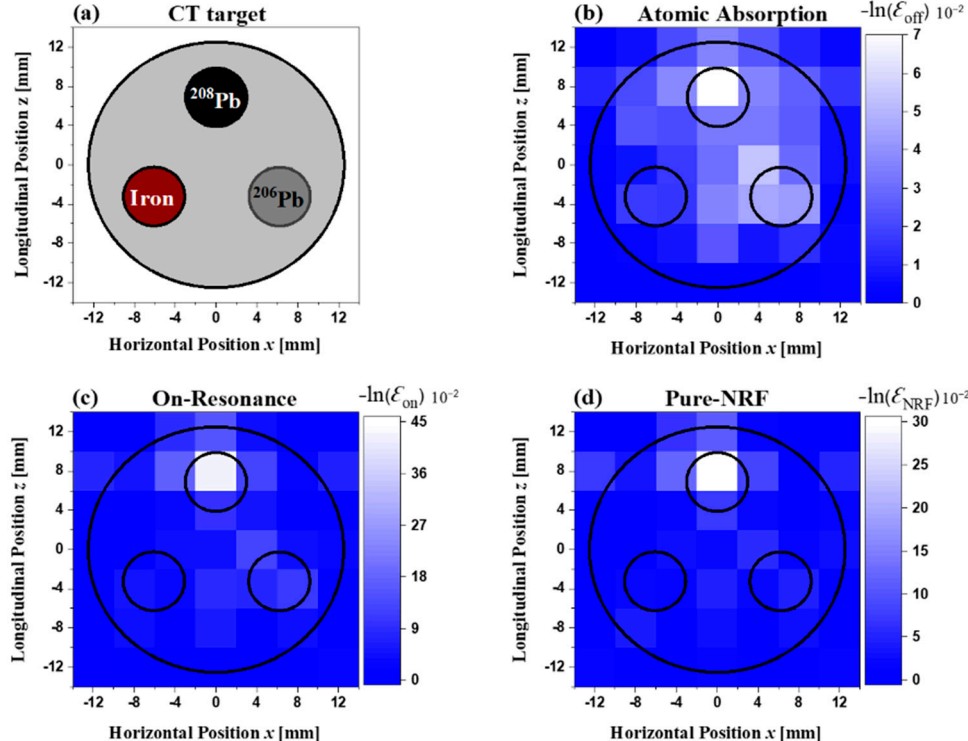

**Figure 8.** (**a**) CT target cross-sectional slice image at *z* = 3 mm, as well as reconstructed CT images of the (**b**) atomic attenuation, (**c**) on-resonance attenuation, and (**d**) pure NRF case.

Figure 9a shows the CT target arrangement at *z* = 11 mm. This figure shows an enriched lead isotope rod ($^{208}$Pb), iron rod, and hollow aluminum rod inserted into the first, second, and third holes, respectively, of the aluminum holder. The CT reconstructed image of the off-resonance attenuation, measured by the LaBr$_3$(Ce) detector, is shown in Figure 9b. A high-attenuation area, induced by the atomic process corresponding to the $^{208}$Pb rod, appeared clearly in the reconstructed image. Because the atomic attenuation that was caused by the iron rod or the hollow aluminum rod was much smaller compared with that caused by the lead rod, the former two rods disappeared in the reconstructed image. Similarly, in the case of the layer at *z* = 3 mm, only the lead isotope rod ($^{208}$Pb) was visible in the on-resonance reconstructed image (Figure 9c) and pure NRF image (Figure 9d), whereas the iron and aluminum rods vanished almost completely.

Figure 10a shows a cross-sectional slice image of the CT target at *z* = 17 mm. This figure shows that two enriched lead isotope rods ($^{206}$Pb and $^{208}$Pb) were inserted into two holes of the aluminum holder, whereas the third hole seems empty. The CT reconstructed image of the off-resonance attenuation, measured by the LaBr$_3$(Ce) detector, is shown in Figure 10b. The two high-attenuation areas, induced by the atomic process corresponding to the positions of the lead isotope rods ($^{208}$Pb and $^{206}$Pb), were clearly visible, whereas the low intensity at the third hole position indicated an empty area. Figure 10c indicates the distribution of $\varepsilon_{off}$. Although the $^{208}$Pb was clearly visible, the $^{206}$Pb rod appeared blurred. Figure 10d shows the distribution of the NRF attenuation caused by $^{208}$Pb, namely pure NRF. Although the $^{208}$Pb rod was clearly visible in the reconstructed image, the $^{206}$Pb rod and empty area almost vanished.

As in the previous study [2], these results show that the NRF-CT method can clearly distinguish the two enriched lead isotope rods ($^{208}$Pb and $^{206}$Pb) within the same sample with isotope-selective capability. A 3D-CT image can essentially be constructed from 2D-CT images measured at different axial positions. In this study, three measured 2D NRF-CT images were used to construct one 3D NRF-CT image. We used the MicroAVS data visualization tool [29] to visualize the 3D NRF-CT images with a 4 mm pixel size in the horizontal plane.

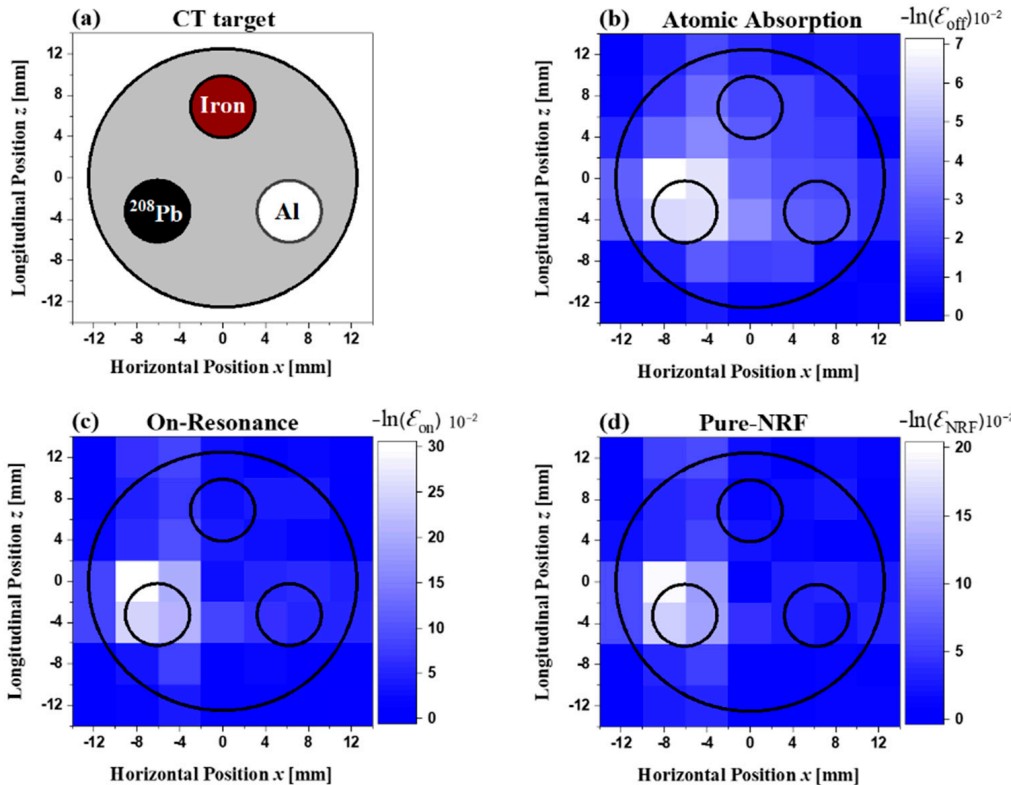

**Figure 9.** (**a**) CT target cross-sectional slice image at *z* = 11 mm, with reconstructed CT images of the (**b**) atomic attenuation, (**c**) on-resonance attenuation, and (**d**) pure NRF case.

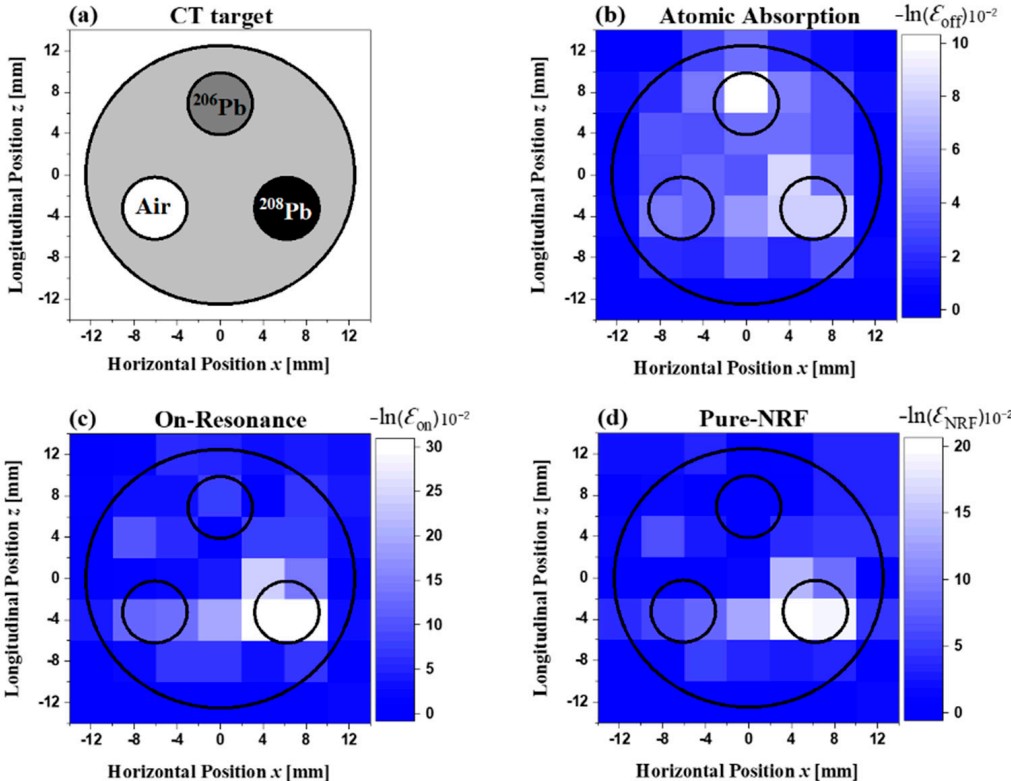

**Figure 10.** (**a**) CT target cross-sectional slice image at *z* = 17 mm, with reconstructed CT images of the (**b**) atomic attenuation, (**c**) on-resonance attenuation, and (**d**) pure NRF case.

Figure 11a–i shows consecutive shots of the three-dimensional reconstructed image of the off-resonance attenuation (i.e., atomic attenuation used in normal gamma ray CT), which were captured from the visualized 3D movie (see video S1 in the supplementary information). These figures clearly show five high-attenuation areas caused by the atomic process, corresponding to the positions of the five lead isotope rods ($^{208}$Pb and $^{206}$Pb) within the CT target. In contrast, the atomic attenuation induced by the iron rod, aluminum rods, or the empty area were almost negligible, and thus it was difficult to identify their locations.

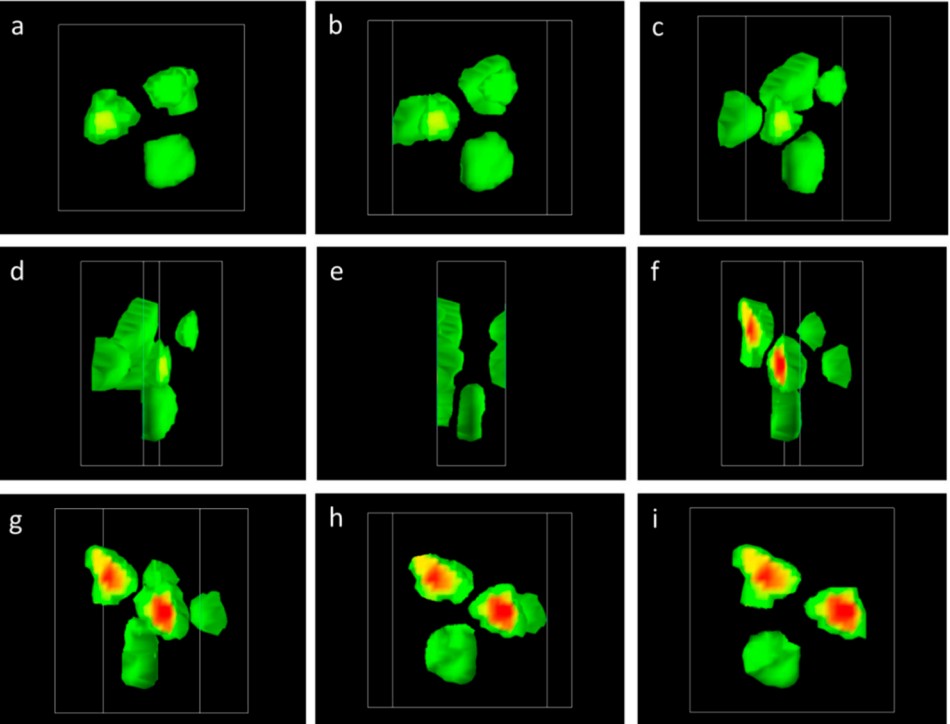

**Figure 11.** (**a–i**) Consecutive shots of the reconstructed image of the off-resonant attenuation (gamma-CT image) captured from the visualized three-dimensional movie.

Figure 12a–i shows consecutive shots of the visualized reconstructed image of the NRF attenuation caused only by the lead isotope rods ($^{208}$Pb, pure NRF) in three dimensions. These figures clearly show the locations of the three enriched lead isotope rods ($^{208}$Pb) in the three *z* layers. In contrast, $^{206}$Pb, the iron and aluminum rods, and the empty areas were invisible (see video S2 in the supplementary information).

In the current study, we demonstrated isotope-selective CT imaging based on NRF in three-dimensional measurements. A measurement time of approximately 48 h was required for obtaining a single 3D NRF-CT image of the examined sample with a 4 mm pixel resolution in the horizontal plane for three vertical positions. Although the resolution of the 3D NRF-CT image in the current study (4 mm pixel size) was lower than the resolution of the 2D NRF-CT which we obtained in the previous study (2 mm pixel size) [2], the improvements in the experimental set-up were positively reflected in the image acquisition time, whereas the current 3D NRF-CT image with three layers in the *z* direction required a shorter acquisition time compared with the 2D NRF-CT image [2]. One of the objectives of this study was to look at the possibility of applying the NRF-CT technique in three dimensions using the current experimental setup of the UVSOR-III synchrotron radiation facility. Consequently, the lack of the obtained image resolution was not proven to be an impediment so far.

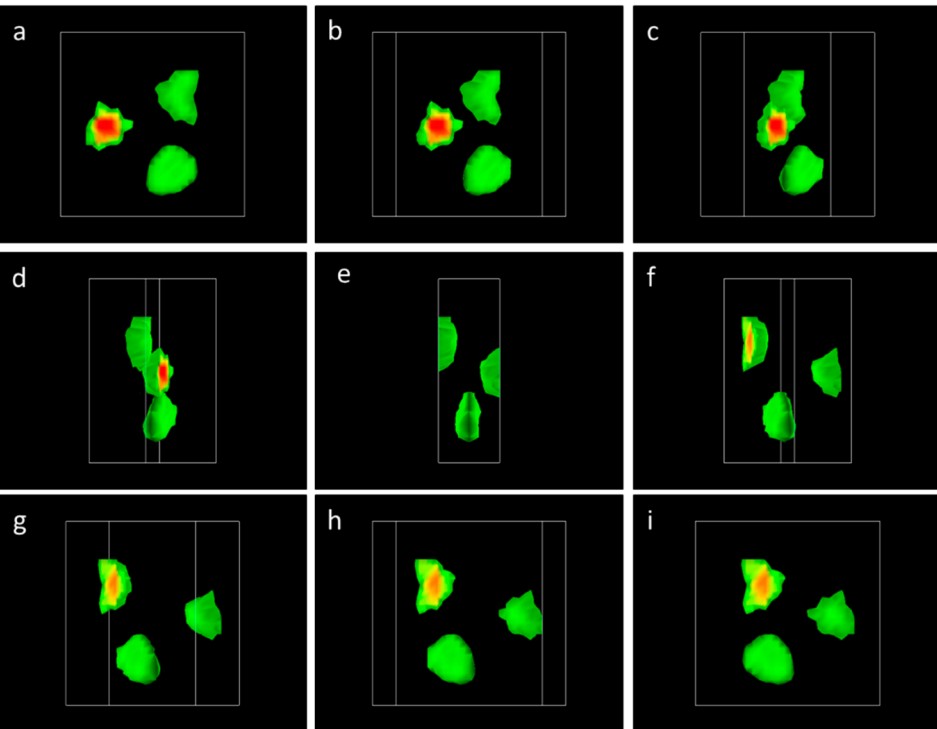

**Figure 12.** (**a–i**) Consecutive shots of the reconstructed image of the nuclear resonant attenuation (pure NRF) captured from the visualized three-dimensional movie.

We agree that the 3D NRF-CT imaging method is still time-consuming. However, some solutions have been proposed to enhance the obtained image quality and reduce the measurement time. Increasing the number of HPGe detectors to measure the NRF gamma rays from the witness target is one of the possible solutions. Another solution is to increase the gamma ray beam intensity. The Extreme Light Infrastructure Nuclear Physics (ELI-NP) project is currently being constructed as a new LCS gamma ray facility in Romania, where the expected flux density of the LCS gamma ray beam (5000 photons$\cdot$s$^{-1}\cdot$eV$^{-1}$) [30] will be 500 times higher than the present one. The development of a new laser-based inverse Compton scattering gamma beam system at the ELI-NP project, which features extremely high intensities at very narrow bandwidths, opens new and important opportunities in nuclear science research and for 3D NRF-CT applications. In addition, there are several proposals to improve the image quality with fused visualization methods [31–35] which combines two different images, such as a three-dimensional ultrasonic image and an X-ray CT image, into a single image.

Presenting the NRF-CT technique in three dimensions for the first time leads to the widespread usage of that technique in the nuclear safeguards. Furthermore, it helps to propose the quantification of hidden isotopes within the SNMs, especially by improving the obtained image quality.

## 4. Conclusions

We experimentally obtained a three-dimensional, isotope-selective CT image of an enriched lead isotope ($^{208}$Pb) distribution, concealed inside a cylindrical aluminum holder together with another enriched lead isotope ($^{206}$Pb), iron and aluminum rods, and vacant areas. We used an LCS gamma ray beam with a maximum energy of 5.528 MeV, a 2 mm beam size, and a total flux density of 10 photons$\cdot$s$^{-1}\cdot$eV$^{-1}$ to excite the $J^{\pi} = 1^{-}$ state at 5.512 MeV in $^{208}$Pb. For each row of the rods, we obtained two-dimensional CT images based on atomic attenuation, NRF attenuation, and the sum of them with a 4 mm pixel size. In the atomic attenuation images, the enriched lead isotope rods ($^{208}$Pb and $^{206}$Pb) were observed. In the NRF attenuation images, only the $^{208}$Pb rods were visible. The corresponding 3D

NRF-CT image showed the isotope distribution of $^{208}$Pb with 4 and 8 mm pixel sizes in the horizontal and vertical planes, respectively. The total required time to obtain a 3D isotope-selective image was 48 h. Further improvements in the image quality as well as a reduction in the acquisition time will be feasible by increasing the number of detectors to measure NRF gamma rays, increasing the gamma ray beam intensity and developing advanced data processing techniques, which will be a significant advancement in nuclear engineering for using the 3D NRF-CT technique in real applications.

**Supplementary Materials:** The following sub-elementary materials are available online at https://www.mdpi.com/article/10.3390/app11083415/s1. Video S1: The three-dimensional movie of the off-resonant attenuation (Gamma-CT image). Video S2: The three-dimensional movie of the nuclear resonant attenuation (Pure NRF image).

**Author Contributions:** Conceptualization, H.O.; Data curation, K.A. and H.Z.; Formal analysis, K.A.; Funding acquisition, H.O., T.H. and T.S.; Investigation, K.A., H.O. and H.Z.; Methodology, All authors; Project administration, H.O.; Resources, H.O., T.H., T.S., M.K.; Software, K.A., H.Z.; Supervision, H.O.; Validation, All authors; Visualization, K.A., H.Z., H.O.; Writing—original draft, K.A.; Writing—review & editing, All authors. All authors have read and agreed to the published version of the manuscript.

**Funding:** This research was funded by the Japan Society for the Promotion of Science (JSPS) KAKENHI under grants 18H01916, 18H03715, and 17K05482.

**Acknowledgments:** This work was performed at the beamline BL1U of the UVSOR-III Synchrotron Radiation Facility with the approval of the Institute for Molecular Science (IMS), National Institute of Natural Science (NINS), Okazaki 444-8585, Japan. (Proposal No. 19-503 and 20-703).

**Conflicts of Interest:** The authors declare no conflict of interest. The funders had no role in the design of the study; in the collection, analyses, or interpretation of data; in the writing of the manuscript, or in the decision to publish the results.

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
