# Peer review of "Three-Dimensional Nondestructive Isotope-Selective Tomographic Imaging of 208Pb Distribution via Nuclear Resonance Fluorescence"

_applsci, doi:10.3390/app11083415_

Round 1

Reviewer 1 Report

The manuscript entitled "Three-Dimensional Nondestructive Isotope-Selective Tomographic Imaging of 208Pb Distribution via Nuclear Resonance Fluorescence" presents new results from the application of an innovative technique, based on Nuclear Resonance Fluorescence and Computed Tomography, capable to build 3D images of 208Pb samples hidden inside a thich Aluminum container.  The technique guarantees isotope selectivity and reasonable times, despite a further reduction of this latter is welcomed.

The manuscript is clear and well written and in my opinion deserves publication. A number of typos and small improvements are found in my notes, written directly on the top of the original manuscript.

Author Response

Response to the Reviewer 1

First and foremost, we would like to express our grateful to the reviewer 1 for his/her thorough reading of the manuscript. We also thank him/her for appreciating the value of Nuclear Resonance Fluorescence based computed tomography imaging with a quasi-monochromatic Laser Compton Scattering gamma-ray beam for its significant advancement in nuclear engineering by the non-destructive inspection of the hidden isotopic compositions of target materials and the isotope selectivity assessment. We also appreciate his/her positive comments and correction to our paper's English. We agree with almost all their suggestions and we have revised our manuscript accordingly to improve the manuscript's quality. A detailed point-by-point answer to all comments can be found below (reviewers ‘comments in black, our replies in red). We hope the reviewers approve of our responses to their feedback.

Sincerely,

Khaled Ali,

Heishun Zen,

Hideaki Ohgaki,

Toshiteru Kii,

Takehito Hayakawa,

Toshiyuki Shizuma,

Hiroyuki Toyokawa,

Masaki Fujimoto,

Yoshitaka Taira,

Masahiro Katoh.

Point 1 (line 37): Reviewer refers to delete the word “which”.

Response 1: Deleted.

Point 2 (line 63): Reviewer refers to delete the word “which”.

Response 2: Deleted.

Point 3 (line 187): Reviewer refers to add the word “are found” to the caption of figure 4.

Response 3: The word was added.

Point 4 (line 310): Reviewer refers to delete the word “ However”.

Response 4: Deleted, and the two sentences were joined together as follows:

Old phrase:

Although the resolution of the 3D NRF-CT image in the current study (4 mm pixel size) is lower than the resolution of the 2D NRF-CT which we obtained in the previous study (2 mm pixel size) [2]. However, the improvements in the experimental set-up were positively reflected on the image acquisition time.

New phrase:

Although the resolution of the 3D NRF-CT image in the current study (4 mm pixel size) is lower than the resolution of the 2D NRF-CT which we obtained in the previous study (2 mm pixel size) [2], the improvements in the experimental set-up were positively reflected on the image acquisition time.

Point 5 (line 311): The Reviewer refers to replace the word “which” by the word “indeed”.

Response 5: The sentence “the current 3D NRF-CT image with three layers in z direction ….” is related to the preceding sentence. Therefore, the word "Whereas" is appropriate for linking the two sentences. We think that using the word "indeed" lets the sentence stand alone, which is not the intended meaning.

Point 6 (line 312): The Reviewer refers to replace the sentence “acquisition time less than that was necessary to obtain one 2D NRF-CT image” by the sentence “a shorter acquisition time compared to the 2D NRF-CT image”.

Response 6: Done.

Point 7 (line 313): The Reviewer refers to replace the sentence “one of the study’s objectives” by the sentence “one of the objectives of the study”.

Response 7: Done.

Point 8 (line 314): The Reviewer refers to delete the word “facility’s”.

Response 8: We think re-phrasing of the sentence tends to be much easier than deleting the word “facility’s” as follows:

Old phrase:

“One of the study’s objectives is to look at the possibility of applying the NRF-CT technique in three dimensions using the UVSOR-III synchrotron radiation facility’s current experimental setup.”.

New phrase:

“One of the objectives of the study is to look at the possibility of applying the NRF-CT technique in three dimensions using the current experimental setup of the UVSOR-III synchrotron radiation facility.”.

Point 9 (line 315): The Reviewer refers to start the word “consequently” with capital letter.

Response 9: Done.

Point 10 (line 320): The Reviewer refers to delete the word “such”.

Response 10: Done.

Point 11 (line 327): The Reviewer refers to replace the word “Moreover” by the word “and”.

Response 11: Done. And the two sentences were joined together as follows:

Old phrase:

“The development of a new laser-based Inverse Compton Scattering gamma beam system at ELI-NP, which features extremely high intensities at very narrow bandwidths, opens new and important opportunities in nuclear science research. Moreover, for the 3D NRF–CT applications.”.

New phrase:

“The development of a new laser-based Inverse Compton Scattering gamma beam system at ELI-NP, which features extremely high intensities at very narrow bandwidths, opens new and important opportunities in nuclear science research and for the 3D NRF–CT applications.”.

Point 11 (line 333): The Reviewer refers to replace the word “with the” by “by”.

Response 11: Done.

Reviewer 2 Report

The paper of Ali et al. "Three-dimensional nondestructive isotope selective tomographic imaging of 208Pb distribution via Nuclear Resonance Fluorescence" describes a next step which this team is making towards the development of the NRF CT technique. 

I find this paper interesting and worth being published. I have few questions: 

(1) How were chosen the steps in the (x,z) and for rotation? Are these the optimal steps? How it relates to the dimensions of the object which is been analyzed and to the beam-spot dimensions?  

(2) When describing the procedure for calculation of the transmission factors, will be much easier to understand if a formula is provided.

(3) In Fig. 6 you need to indicate the 1.461 KeV peak in the spectra and discuss what are the other peaks, which are seen. 

(4) When discussing the algebraic reconstruction technique, it is not enough to refer to the paper of Zen et al., but rather briefly summarize the steps which are undertaken. 

Author Response

Response to the Reviewer 2

We would like to thank reviewer 2 for appreciating the value of Nuclear Resonance Fluorescence-based computed tomography imaging with a quasi-monochromatic Laser Compton Scattering gamma-ray beam for its significant advancement in nuclear engineering by the non-destructive inspection of the hidden isotopic compositions of target materials and the isotope selectivity assessment. We appreciate his/her insightful suggestions and efforts toward enhancing our manuscript, as well as the supportive and positive reviews. We have integrated the suggestions into the manuscript to strengthen and clarify it. A comprehensive point-by-point response to all comments is given below (reviewers' comments are in black, and our responses are in red).

Sincerely,

Khaled Ali,

Heishun Zen,

Hideaki Ohgaki,

Toshiteru Kii,

Takehito Hayakawa,

Toshiyuki Shizuma,

Hiroyuki Toyokawa,

Masaki Fujimoto,

Yoshitaka Taira,

Masahiro Katoh.

Point 1: How were chosen the steps in the (x, z) and for rotation? Are these the optimal steps? How it relates to the dimensions of the object which is been analyzed and to the beam-spot dimensions?

Response 1:

We chose the smallest number of the measurement positions that would allow us to obtain a reasonable image in an appropriate time frame and to prove the NRF nondestructive analysis in three dimensions using the LCS gamma-ray beam. The step sizes in individual directions were determined to obtain a reasonable image resolution in considering the CT target geometry under limited measurement time. The diameter of the gamma-ray beam is 2 mm, which is half the pixel size of the reconstructed image. The CT target had a diameter of 25 mm, and the lead and iron rods in the target had a diameter of 6 mm. The gamma-ray beam diameter was sufficiently shorter than the object being studied.

We added this description to the manuscript.

Point 2: When describing the procedure for calculation of the transmission factors, will be much easier to understand if a formula is provided.

Response 2: We added a brief description of the transmission factors equations to the main text of the result section as follows:

(Please find the attached response letter to see the inserted equations). The equations parameters definitions were included in the manuscript.

Point 3: In Fig. 6 you need to indicate the 1.461 KeV peak in the spectra and discuss what are the other peaks, which are seen.

Response 3: The appeared peaks of the spectra in figure 6 will be clarified in the following:

  • 1460 keV of 40K (Natural isotope).
  • 2616 keV of 208Tl (Natural isotope).
  • 4490 keV of the double escape peak of the NRF peak of 208Pb at 5512 keV.
  • 5001 keV of the single escape peak of the NRF peak of 208Pb at 5512 keV.
  • 5292 keV and 5512 keV are the NRF levels of 208

We modified fig. 6 to display the peaks that were appeared on the spectrum. In addition, we modified the description of fig. 6 in the main text to clarify them.

Point 4: When discussing the algebraic reconstruction technique, it is not enough to refer to the paper of Zen et al., but rather briefly summarize the steps which are undertaken.

Response 4: We briefly summarized the ART steps in the result section as follows:

In the NRF-CT imaging, several CT reconstruction algorithms could be used. Due to the limited number of the measured projections in the current study, one of the iterative algorithms called the algebraic reconstruction technique (ART) [26–28] was used. A program has been developed in LabVIEW to reconstruct the images. The ART is a sequential approximation method for the image reconstruction from a series of angular projections (sinogram). To perform the ART algorithm, the sinogram data points are to be processed in a series of steps. We created a two-dimensional primary image as a two-dimensional matrix with zero in every cells. The cell number of the matrix are determined based on the projection numbers obtained during the experiment. The primary matrix in our case will be 7×7. To mitigate the errors of the matrix rotation caused by small matrix size of sinogram, pre- and post-process have been implemented. The difference between the projections calculated from the reconstructed image and the measured projections is computed and used to correct the reconstructed image iteratively. The convergence of the reconstruction is evaluated by the root-mean-square-difference between the previous and current reconstructed images. More details of the pre- and post-process are provided in the report [15].

Reviewer 3 Report

The conclusions could be improved and presented more concisely

Author Response

Response to the Reviewer 3

We would like to express our appreciation to reviewer 3 for taking the time to read our manuscript thoroughly, and for his/her appreciating to the value of Nuclear Resonance Fluorescence-based computed tomography imaging with a quasi-monochromatic Laser Compton Scattering gamma-ray beam for its significant advancement in nuclear engineering by the non-destructive inspection of the hidden isotopic compositions of target materials and the isotope selectivity assessment. He/she is generally concerned that our original manuscript's conclusion could be improved and presented more succinctly. We make every attempt to solve these issues. We hope the reviewer 3 approve of our responses to their feedback.

Sincerely,

Khaled Ali,

Heishun Zen,

Hideaki Ohgaki,

Toshiteru Kii,

Takehito Hayakawa,

Toshiyuki Shizuma,

Hiroyuki Toyokawa,

Masaki Fujimoto,

Yoshitaka Taira,

Masahiro Katoh.

Point 1: The conclusions could be improved and presented more concisely

Response 1: We have modified the conclusion to be more concisely as follows:

The new conclusion:

“We experimentally obtained a three-dimensional isotope-selective CT image of an enriched lead isotope (208Pb) distribution, concealed inside a cylindrical aluminum holder together with another enriched lead isotope (206Pb), iron and aluminum rods, and vacant areas. We used an LCS gamma-ray beam with a maximum energy of 5.528 MeV, 2 mm beam size, and a total flux density of 10 photons·s-1·eV-1 to excite the Jπ =1- state at 5.512 MeV in 208Pb. For each row of the rods, we obtained two dimensional CT images based on atomic attenuation, NRF attenuation and sum of them with 4-mm pixel size. In the atomic attenuation images, the enriched lead isotope rods (208Pb and 206Pb) were observed. In the NRF attenuation images, only the 208Pb rods were visible. The corresponding 3D NRF–CT image showed the isotope distribution of 208Pb with 4- and 8-mm pixel sizes in the horizontal and vertical planes, respectively. Total required time to obtain a 3D isotope-selective image was 48 hours. Further improvements in the image quality as well as reduction in the acquisition time will be feasible by increasing the number of detectors to measure NRF gamma rays, increasing the gamma-ray beam intensity, and developing advanced data processing techniques, which will be a significant advancement in nuclear engineering to use the 3D NRF-CT technique in the real applications.”.
